# A COMPUTATIONAL APPROACH TO VISUAL ECOLOGY WITH DEEP REINFORCEMENT LEARNING

## ABSTRACT

Animal vision is thought to optimize various objectives from metabolic efficiency to discrimination performance, yet its ultimate objective is to facilitate the survival of the animal within its ecological niche. However, modeling animal behavior in complex environments has been challenging. To study how environments shape and constrain visual processing, we developed a deep reinforcement learning framework in which an agent moves through a 3-d environment that it perceives through a vision model, where its only goal is to survive. Within this framework we developed a foraging task where the agent must gather food that sustains it, and avoid food that harms it. We first established that the complexity of the vision model required for survival on this task scaled with the variety and visual complexity of the food in the environment. Moreover, we showed that a recurrent network architecture was necessary to fully exploit complex vision models on the most visually demanding tasks. Finally, we showed how different network architectures learned distinct representations of the environment and task, and lead the agent to exhibit distinct behavioural strategies. In summary, this paper lays the foundation for a computational approach to visual ecology, provides extensive benchmarks for future work, and demonstrates how representations and behaviour emerge from an agent's drive for survival.

## 1 INTRODUCTION

Successful theories of sensory neural circuits often begin with a careful choice of objective function. For example, efficient coding posits that sensory systems minimize the redundancy of the neural code, and explains how sensory circuits maximize information capacity at minimal metabolic cost (Barlow, 1961; Olshausen & Field, 2004). Predictive coding and the Bayesian brain hypothesis assume that neural systems aim to approximate optimal Bayesian inference, and explain phenomena ranging from sensory circuit organization up to decision making and behaviour (Rao & Ballard, 1999; Beck et al., 2012; Pouget et al., 2013; Millidge et al., 2022). Finally, objective functions such as natural image classification lead artificial neural networks to exhibit similar patterns of activity as sensory neural circuits (Yamins & DiCarlo, 2016; Richards et al., 2019).

However, the ultimate objective of any neural circuit is to facilitate the survival of an animal within its ecological niche. Although the aforementioned objective functions provide invaluable perspectives on sensory coding, we cannot develop a complete understanding of animal sensory processing without accounting for the niche in which they evolve, and the behavioural strategies this niche requires (Baden et al., 2020). This, of course, is easier said than done, as we lack computational tools for effectively modelling the incredible complexity of an ecological niche.

To address this need, we developed a deep reinforcement learning (RL) framework for studying visual and sensory ecology. In our framework we reduced the reward function to the survival of the agent, and avoided further fine tuning of the reward. We then studied the average lifespan of agents, as well as the representations and behaviours that result from particular combinations of vision model, overall "brain" model, and environment. A key advantage of our deep RL approach is that the vision and brain models of the agent take the form of conventional artificial neural networks (ANN). As such, established ANN models of neural circuits can easily be imported into our framework.

We used the RL environment to model a foraging task, where the agent had to gather food that sustained it and avoid food that harmed it. We then characterized how the complexity of the vision

model scaled with the visual complexity of the food that the agent had to recognize to ensure its survival. We established that a linear vision model is sufficient for survival when there are only two visually distinct food sources, but that as we increased the visual complexity of the food sources — up to each food source being visualized by a class of CIFAR-10 images — more complex vision models were required to maximize agent survival.

We then compared various architectures of brain model of the agent, including feedforward and recurrent models, and models that received readouts of the metabolic state of the agent. We found that these architectural variations not only affected agent survival, but also qualitatively altered the behaviour and representations of the agent. One one hand, we found that providing metabolic readouts to the agent helped it to avoid overeating and depleting food in the environment. On the other, we found that recurrence both facilitated object discrimination on visually complex tasks, and allowed it to capture features of the environment beyond the immediate presence of food.

## 1.1 BACKGROUND AND RELATED WORK

Efficient use of computational resources is a major challenge in deep RL, and the IMPALA RL architecture (Espeholt et al., 2018) laid the groundwork for many approaches and studies. This includes the "For The Win" (FTW) architecture, which successfully trained agents to achieve human-level playing in first-person environments (Jaderberg et al., 2019), and the work of Merel et al. (2019), where they trained an agent with a detailed mouse body to solve naturalistic tasks.

ANNs have long served as models of sensory neural circuits. Recent papers have sought to mimic the architecture of animal visual systems with convolutional neural networks (CNNs), and model features of early vision by training the CNNs on naturalistic stimuli (Lindsey et al., 2018; Qiu et al., 2021; Maheswaranathan et al., 2023).

Our work builds on these two streams of research. In particular, we explored how agents survive in naturalistic, first-person environments that necessitate precisely-tuned behaviours, as well as vision systems with high acuity.

## 2 COMPUTATIONAL FRAMEWORK

To simulate visual environments we relied on *ViZDoom* Schulze & Schulze (2019), an RL-focused, high-throughput simulation engine based on the computer game *Doom*, that simulates pseudo 3-d environments, and provides agents with rich visual input. In general, our environments were 2-d planes populated with different objects (Fig. 1**a**). The environment had the base appearance of a grassy field and a blue sky, and the objects within it were visualized by various textures depending on the task. The agent perceived the environment through a $160 \times 120$ pixel, RGB viewport. We measured environment time in simulation frames — when rendered for human perception, simulations are intended to run at 35 frames per second (FPS), so that 1000 frames is a little under 30 seconds of real-time.

## 2.1 VISION AND BRAIN MODEL ARCHITECTURE

Every fourth frame in the simulation was processed by the vision model of the agent, which is defined as a CNN implemented in *PyTorch* (Fig. 1**b**). We modeled the CNN after the early mammalian visual system: the base layers were grouped sequentially into the photoreceptor (PR), bipolar (BP), retinal ganglion cell (RGC), lateral geniculate nucleus (LGN), and primary visual cortex (V1) layers. Each group comprised a convolutional layer with a certain number of channels (NC) and kernel size (KS), followed by the ELU nonlinearity, followed by a pooling layer with non-overlapping windows of size KS (Tab. 1). The number of channels in each layer was governed by $n_{BC}$ (number of base channels), except at the LGN where it was governed by $n_{LGN}$.

The output of the CNN was fed into a sequence of three fully connected (FC) layers each with $n_{FC}$ neurons and ELU nonlinearities. Some models that we considered

| Class | NC | KS | Pooling |
|-------|-----|-----|---------|
| PR | 3 | — | — |
| BP | $n_{BC}$ | 3 | Average |
| RGC | $2n_{BC}$ | 3 | Average |
| LGN | $n_{LGN}$ | 1 | — |
| V1 | $4n_{BC}$ | 4 | Max |

Table 1: Vision Model

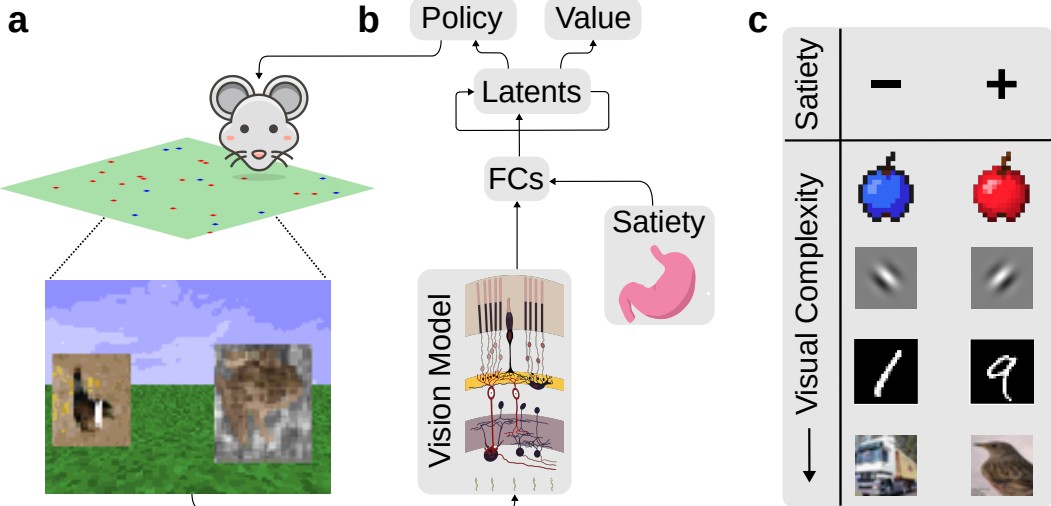

Figure 1: *Overview of the computational framework.* **a**: A depiction (top) of the agent (mouse) in an environment (green field), surrounded by nourishment (red dots) and poison (blue dots). The agent perceives the environment through its viewport (bottom). **b**: The brain model of the agent processes the viewport with a CNN modelled after early animal vision (bottom). The output of the vision model and the satiety signal feed into the FC layers which can also be modulated by input satiety, and which ultimately outputs into a GRU. The policy and estimated value function (top) are linear functions of the latent, GRU state. **c**: Task complexity varies with visual complexity, from food represented by apples to CIFAR-10 images.

also include what we call input satiety (IS), where we added an extra input neuron to the second FC layer of the network that equaled the current satiety of the agent (we formally explain satiety in the next subsection). For recurrent (RNN) brain models, the FC layers fed into a gated recurrent unit (GRU) with $n_{FC}$ neurons. For feedforward (FF) brains, the FC layers fed into another FC layer with $n_{FC}$ neurons and a sigmoid nonlinearity. We consider both feedforward (FF-IS) and recurrent (RNN-IS) models with input satiety, and in all cases we refer to activity in the output layers as the latent state of the agent.

The policy of our agent was represented by two, independent categorical distributions: the heading distribution over the actions left, right, and centre, and the velocity distribution over the actions forward, backward, and stationary. We used an actor-critic architecture for our agents, and both the actor and critic were linear functions of the latent state. We trained our agents with proximal policy optimization (PPO) (Schulman et al., 2017) using the *Sample Factory* Petrenko et al. (2020) library, which provided a high-performance, asynchronous implementation of PPO.

## 2.2 THE FORAGING TASK

In our foraging task the agent had a satiety (inverse hunger) state that ranged from 0 to 100, and the reward at every frame was equal to the current satiety of the agent. The satiety of the agent decreased at a constant rate, and the agent died/the simulation ended when satiety reached 0. Every environment was populated with 10 classes of objects, which added -25, -20, -15, -10, -5, 10, 20, 30, 40, or 50 to satiety, respectively, depending on the class, up to the bound of 100 on satiety. The environment was initialized with a greater abundance of positive objects (nourishment) than negative objects (poison), and after environment initialization new nourishment and poison objects were generated at the same, constant rate. A consequence of this design was that the task tended to become more difficult over time as the agent consumed nourishment and thereby lowered the ratio of nourishment to poison.

The primary way we varied task difficulty was with the textures that represented each class (Fig. 1**c**). The four tasks we considered were: The *apples task*, where the positive and negative objects classes were represented by red and blue apples, respectively. The *Gabors task*, where each object class was

represented by one of eight Gabor patches, and in particular -5 and -10, and 10 and 20 each share a patch texture. The *MNIST task*, where each object class was associated with a digit, and each object was represented by a randomly chosen image from MNIST for the associated digit. Finally, the *CIFAR-10 task*, where each object class was associated with a CIFAR-10 class.

## 3 RESULTS

One of our aims was to provide a broad spectrum of benchmarks for studying synthetic ecology within an RL setting. We ran our simulations on a high performance computing cluster, where we had access to 18 Nvidia A100 GPUs and 4 AMD EPYC 7742 CPUs. We trained most models for $8 \cdot 10^9$ simulation frames each — on our hardware we achieved training speeds of around 40,000 frames per second, so that the average wall-clock training time for a single model was around 2 and a half days. Running all the training simulations that we present took approximately 2 months using all available hardware.

The standard hyperparameters for the recurrent neural networks (RNN or RNN-IS) were $n_{BC} = 16$, $n_{LGN} = 32$, and $n_{FC} = 128$; for the feedforward networks (FF or FF-IS) $n_{FC} = 32$ and all other parameters remained the same. We always state any deviations from this standard. For all combination of hyperparameters we repeated training simulations three times, and throughout the paper we typically report the median, minimum, and maximum of the quantities of interest (e.g. survival time).

### 3.1 VISION MODEL COMPLEXITY SCALES WITH TASK DIFFICULTY

We began by characterizing how agent lifespan depended on both the complexity of the vision model and the visual complexity of the objects it had to recognize. We textured our objects with canonical image datasets used in machine vision and vision neuroscience (Fig. 2**a**), so as to guide our expectations of agent performance with established results for ANNs on classification tasks. That being said, our RL agents had to learn to solve a multi-scale, multi-object discrimination problem based on these textures. As a performance baseline, we note that an agent that took no actions survived for 200 simulation frames.

We first studied how the training time of RNN agents depended on the visual complexity of the task (Fig. 2**b**). We saw that average lifespan on the apples and Gabors tasks converged within the training time of $8 \cdot 10^9$ video frames, yet the MNIST task, and especially the CIFAR-10 task could benefit from additional training time. We ran an additional set of simulations (not shown) on the CIFAR-10 task where we increased training time fivefold to $4 \cdot 10^{10}$ frames, and found that average agent lifespan increased by approximately 20% over the model trained for $8 \cdot 10^9$ frames, and that this amount of training was sufficient for convergence. We also saw that learning exhibits a high degree of stability, with little difference between minimum and maximum performance.

We next analyzed how the architecture of the brain model impacted average lifespan (Fig. 2**c**). In particular, we considered linear FF (an FF model with "nonlinearities" given by the identity function), FF, FF-IS, RNN, and RNN-IS models. We saw that while the linear FF architecture was sufficient for achieving relatively good survival times on the apples task, it fared poorly on all other tasks. We found that the nonlinearities in the FF network allowed the agent to achieve significant improvements over the linear network on the Gabors and MNIST tasks, yet the gains were minimal on CIFAR-10. We observed that the RNN architecture facilitated major improvements to lifespan on all tasks, and seemed particular necessary for achieving lifespans of more than a few hundred frames on CIFAR-10. Input satiety also facilitated large performance gains, with the RNN-IS agents achieving the longest lifespans in every case. We also noted that the more complex brain architectures, especially the IS architectures, achieved higher performance on the Gabors task than the apples task, indicating that they took advantage of the extra information about object nourishment available in the Gabors task.

We then studied how varying the number of base channels $n_{BC}$ and the size of the LGN $n_{LGN}$ in the vision model affected the survival of agents with the FF and RNN brain models (Fig. 2**d-e**). Trends suggested that FF models benefited from fewer parameters on the apples and Gabors tasks, and benefited slightly from additional parameters on the MNIST task, yet lifespans of FF agents on CIFAR-10 remained limited regardless of vision model complexity. The lifespan of RNN agents ex-

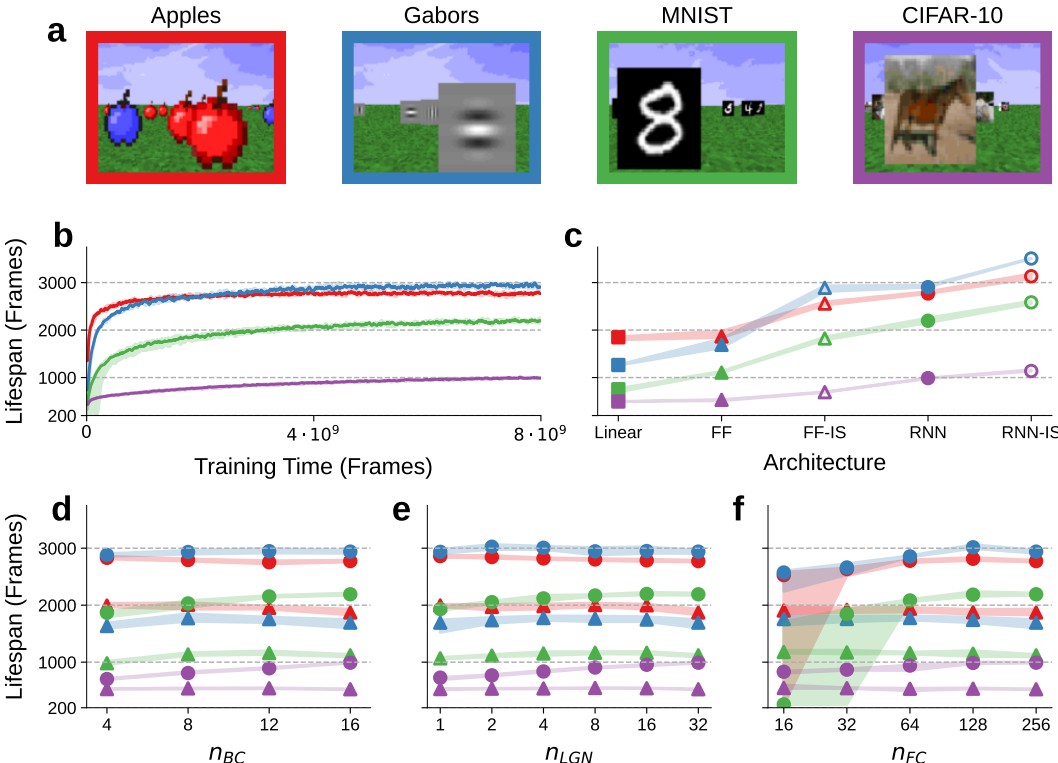

Figure 2: *Benchmarking vision models and brain architectures.* **a**: Sample viewports from the apples (red frame), Gabors (blue frame), MNIST (green frame), and CIFAR-10 task (purple frame). **b**: Median (points) and min-max range (filled area) of 3 training histories of an RNN on each task. Each training history is composed of 10,0000 samples and smoothed with a sliding Gaussian window of size 51. **c**: Average lifespan of the linear (squares), FF (filled triangles), FF-IS (empty triangles), RNN (filled circles), and RNN-IS (empty circles) brains. Lifespan computed as the average over the last 500 steps of the smoothed training histories. **d**: Lifespan of FF (triangles) and RNN agents (circles) as a function of base channels $n_{BC}$. **e**: Lifespan of the FF and RNN agents as a function of LGN size $n_{LGN}$. **f**: Lifespan of the FF and RNN as a function of latent space size $n_{FC}$.

hibited similar trends, except on the CIFAR-10 task where RNN agents revealed the largest positive correlation between vision model complexity and lifespan of any architecture and task combination. We also modulated the size of the FC and latent layers $n_{FC}$ (Fig. 2**f**), and found that while FF agent survival was insensitive to $n_{FC}$, RNN agents profited from larger values for $n_{FC}$ on all tasks. We also noted that RNN agents would sometimes fail to learn the task at all when trained with smaller values of $n_{FC}$ (indicated by the shaded min-max region reaching the baseline of 200) — in these cases agents learned a policy of taking no actions.

### 3.2 RECURRENCE FACILITATES DISCRIMINATION OF VISUALLY COMPLEX OBJECTS

In order to disentangle lifespan increases due to better object recognition from increases due to smarter behaviour, we characterized the discrimination performance of the FF, FF-IS, RNN, and RNN-IS brain architectures by counting the relative frequencies with which the agent picked up each poison and nourishment (Fig. 3). We noted that as a performance baseline, agents with no ability to discriminate between objects would pick up poison or nourishment with an equal frequency of 0.1.

On the apples and the Gabors task we found that all architectures performed equally well, and sought nourishment and avoided poison with similar frequencies (Fig. 3**a, b**). Although agents easily distinguished red and blue apples, they were indeed unable to recognize which apples were more or less nourishing (or more or less poisonous; Fig. 3**a**). For the Gabors task on the other hand, agents tended to most strongly avoid and seek the most poisonous and nourishing foods, respectively.

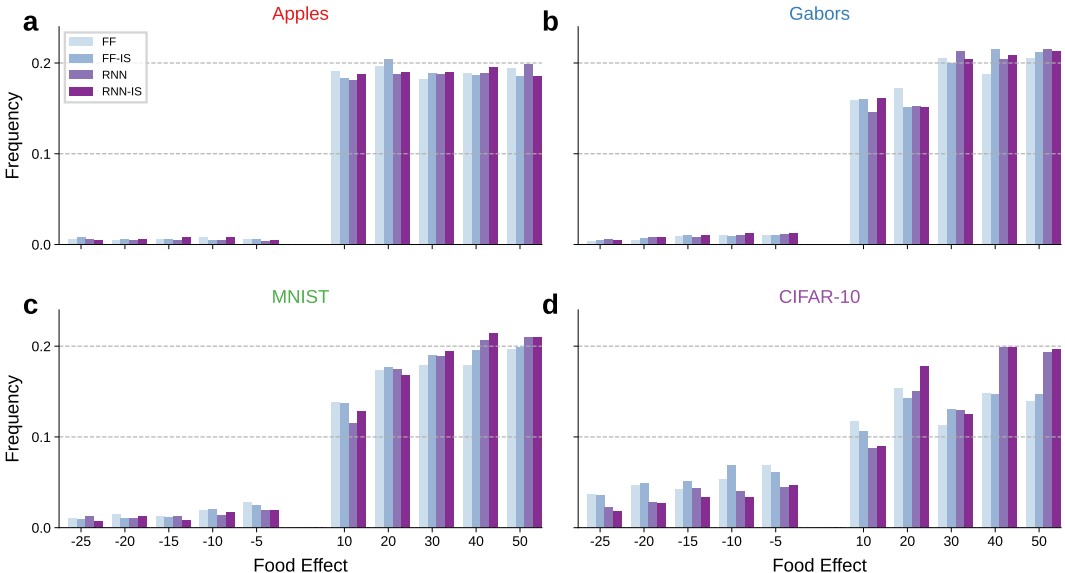

Figure 3: *Characterizing the discrimination performance of different architectures.* Median frequency of pickups of poison and nourishment for the FF (light blue), FF-IS (blue), RNN (purple), and RNN-IS (magenta) agents on **a**: the apples task, **b**: the Gabors task, **c**: the MNIST task, and **d**: the CIFAR-10 task.

We saw similar avoidance and seeking trends on the MNIST task (Fig. 3**c**) as the Gabors task, yet the increased difficulty of the task was reflected in the frequency of most pickups trending closer to 0.1. Moreover, RNN architectures began to demonstrate improved discrimination performance on the MNIST task relative to the FF architectures. Finally, on the CIFAR-10 task (Fig. 3**d**) we observed a large increase in difficulty, and a particularly strong decline in discrimination performance for FF architectures. In all cases, we found little systematic difference in the discrimination performance of agents with or without input satiety.

We thus concluded that a recurrent brain architecture is required to fully exploit complex vision models, and achieve non-trivial lifespans on the CIFAR-10 task. Nevertheless, we saw in the previous subsection that advanced brain architectures led to significant performance improvements regardless of task visual complexity. As such, we also concluded that maximizing lifespan in the foraging environment required not only maximizing discrimination performance, but also implementing better behaviour and strategies.

### 3.3 MODEL BRAIN ARCHITECTURE SHAPES AGENT REPRESENTATIONS OF VALUE

To isolate the source of lifespan gains in the representations learned by the agent, we began by qualitatively examining the estimated value function $\hat{V}$ of various agents, which offers a scalar summary of the agent's perceived quality of its current state. (Fig. 4). We recorded 1000-frame videos of agent behaviour, and studied the sensitivity of $\hat{V}$ to changes in the current input image (Fig. 4**a**) with the method of integrated gradients (Fig. 4**b**, Sundararajan et al. (2017)). We observed that all architectures learned to accurately segment multiple food objects in parallel. We then studied the dynamics of $\hat{V}$ (Fig. 4**c**), and observed that the functions $\hat{V}$ learned by RNN agents were noticeably smoother. We also compared $\hat{V}$ with the agent's satiety, and noticed two key features: firstly, that the magnitude of satiety for architectures appeared predictive of $\hat{V}$ for all architectures, except for FF, and secondly, that the time until a satiety jump (corresponding to food pickups) appeared predictive of $\hat{V}$ for all architectures, but especially FF.

To quantify how important these features were to determining $\hat{V}$, we performed a regression analysis of $\hat{V}$ using 100,000 frames of simulated data (Fig. 5). Because FF architectures appeared to exhibit a degree of task-irrelevant, high-frequency noise (Fig. 4**c**), we first estimated the intrinsic noise of

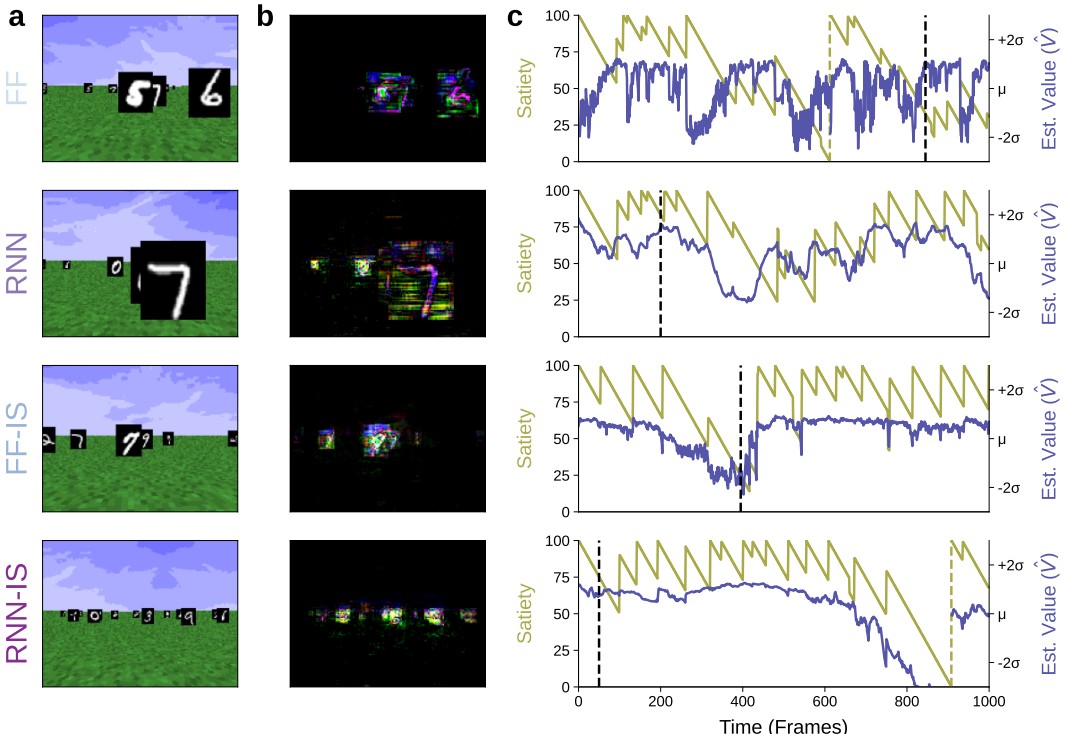

Figure 4: *Qualitative analyses of the estimated value function.* Each row visualizes a 1000 frame simulation of a trained agent on the MNIST task. **a**: Sample viewport from trained models with the FF, FF-IS, RNN, and RNN-IS architectures. **b**: Sensitivity of $\hat{V}$ of each architecture to the corresponding viewports in **a** based on the method of integrated gradients. **c**: The satiety (gold line) and $\hat{V}$ (purple line) of the agent over time. The dashed gold lines indicate a task reset, and the dashed black line indicates the time of the corresponding viewport shown in **a**.

each $\hat{V}$ by convolving it with a sliding window of 20 frames, computing the standard error of the residuals, and using this to indicate an upper bound on the coefficient of determination ($r^2$). We then estimated $\hat{V}$ as a function of agent satiety (Fig. 5**a**), and found that satiety was indeed highly predictive of $\hat{V}$ for IS architectures. Unsurprisingly, satiety was unpredictive of $\hat{V}$ for FF agents, as they had no capacity for estimating their own satiety. On the other hand, the memory afforded to RNN agents allowed them to estimate their own satiety, and satiety was ultimately predictive of $\hat{V}$ for RNN agents.

When we regressed $\hat{V}$ on the time until the next satiety jump (food countdown) alone (Fig. 5**b**), we found it to be highly predictive of $\hat{V}$ for FF architectures, especially in light of the estimated upper bound, yet more limited for other architectures. Finally, when we regressed on both satiety and food countdown (Fig. 5**c**), we found we could explain around 3/4 of the variance in $\hat{V}$ for FF-IS agents, with an even larger fraction for the CIFAR-10 task. Although the variance explained in $\hat{V}$ for FF agents was lower than for FF-IS agents, we presumed that FF agents had no ability to capture additional features that cannot be captured by FF-IS agents, and therefore that most of unexplained variance in $\hat{V}$ for the FF agent was incidental to task performance. In contrast, while satiety and food countdown explained a moderate share of the variance in $\hat{V}$ for the recurrent architectures, we concluded that these value functions were driven by additional task-relevant latent variables.

## 3.4 INPUT SATIETY ENABLES BETTER FORAGING STRATEGIES

In our final analysis we sought to characterize how the behaviour of IS agents allowed them to achieve additional lifespan (Fig. 6). In our recorded videos we noticed that IS agents often paused

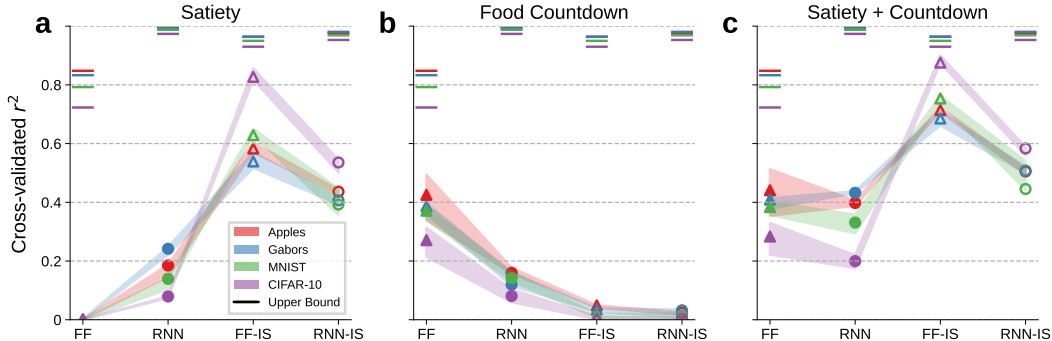

Figure 5: *Regression analysis of the estimated value function.* 10-fold cross-validated $r^2$ for the linear prediction of $\hat{V}$ given distinct regressors. Colours indicate task, points indicate median $r^2$, filled areas indicate min-max $r^2$, and bounding lines indicate performance upper bounds based on the estimated intrinsic noise of $\hat{V}$. **a**: Regression of $\hat{V}$ at time $t$ on agent satiety at time $t$. **a**: Regression of $\hat{V}$ on the time in frames (countdown) until the next food is consumed. **c**: Regression of $\hat{V}$ on both satiety and food countdown.

in front of nourishment before eventually consuming it. To quantify this we computed the average velocity policy over 100,000 simulation frames. We found that IS agents spent more than twice as much time in a stationary state than agents without IS, and when comparing FF-IS to FF, and RNN to RNN-IS architectures, IS architectures spent more time going backwards as well.

In our videos we then noticed that agents only paused when their satiety was already high, suggesting that they were avoiding consuming nourishment that would bring them above 100 satiety, and thus be partially wasted. We therefore computed the percentage of total accumulated nourishment that the agent wasted, and found indeed that IS facilitated a large drop in waste nourishment across tasks and architectures (Fig. 6**b**). Surprisingly, even though we previously showed that RNN agents encode information about their satiety, we found that they were in fact more wasteful than FF architectures. We also observed that the apples task led to the most wasted nourishment, presumably because the agent could not discriminate the degree of nourishment provided by each apple. Overall we found that task difficulty was inversely proportional to wasted nourishment, presumably, at least in part because agents were more hungry whenever they managed to eat.

## 4 DISCUSSION

In this paper, we presented a computational framework studying visual ecology using deep reinforcement learning, and studied the lifespan of agents in a foraging task with visually diverse foods. We systematically explored how the complexity of the agent's vision model interacted with the visual complexity of its environment, and unsurprisingly found that complex vision models facilitated agent survival in more visually complex environments. Yet we also showed that recurrent brain architectures were critical for fully exploiting complex vision models on the most visually demanding tasks, demonstrating how even the perception of static objects benefits from integrating visual information over time. Overall, successful agents developed an impressive ability to segment their visual input, and race towards nourishment to avoid starvation.

In spite of the apparent simplicity of the foraging task, we found that an agent's lifespan did not reduce to its ability to discriminate foods, and that more complex brain architectures improved survival through sophisticated behaviours. In particular, we demonstrated how recurrent brains attended to latent variables beyond the agent's hunger and the immediate presence of food. We also found that a simple signal indicating agent hunger allowed the agent to avoid overeating and achieve superior task performance. Yet even though RNN architectures could estimate hunger, they did not exhibit this behavioural strategy without an explicit hunger signal.

Our framework opens up the way to a rigorous integration of efficient coding theories in computational neuroscience in light of behavioral demands. Towards this goal, we aim to better isolate

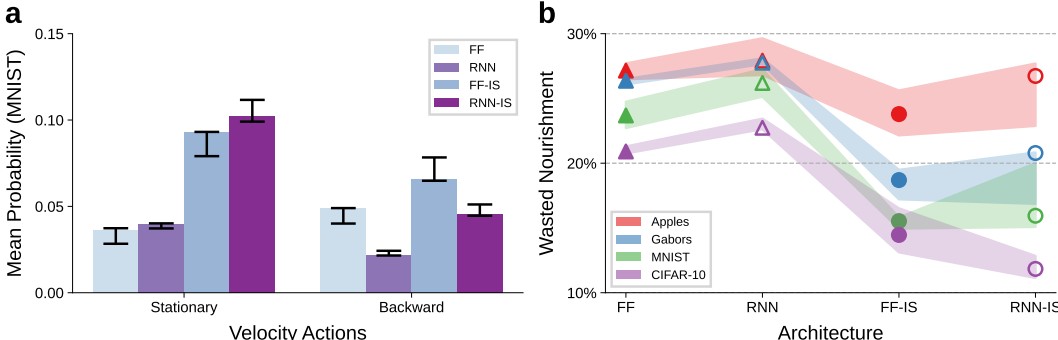

Figure 6: *Characterizing behaviour of agents with input satiety.* **a**: Median (bar height) and min and max (error bars) of the average probability of velocity actions on the MNIST task over 3 simulations of 100,000 frames. **b**: Wasted nourishment as a percentage of total accumulated nourishment per task and architecture.

the environmental variables that drive the latent activity and value functions of our agents using methods for analyzing neural population dynamics (Whiteway & Butts, 2019). Secondly, we aim to better characterize how our RNN agents compressed past sensory information and integrated it with incoming observations, which was critical for survival in our CIFAR-10 task. This process is analogous to Kalman filtering, which is known to be a good model of how cortical areas track low-dimensional variables (Funamizu et al., 2016). Understanding how our agents solve this information integration problem should provide insights into efficient neural coding in high-dimensional, visually rich, dynamic environments.

One behaviour of our agents that we observed yet did not report is that they rarely walked straight towards food, but rather exhibit high-frequency, left and right jitters along their trajectory. We could not rule out that these were simply compensatory movements to not miss the target. Nevertheless, it is striking that visual systems take advantage of retinal jitter to improve visual acuity (Rucci et al., 2007; Intoy & Rucci, 2020), and in future work we aim to identify whether the movement jitters of our agents also improve their visual acuity.

Silver et al. (2021) proposed that sophisticated intelligence could emerge from simple reward functions, and indeed, we believe that understanding the neural architectures and behaviours that facilitate survival requires specifying the details of survival as little as possible. There are, however, major challenges in this approach. Firstly, maximizing simulation throughput is still not sufficient for deep RL architectures to capture many basic behaviours in 3-d environments (Petrenko et al., 2021). Secondly, normative principles such as survival and efficient coding can only take us so far in understanding neural circuits, as many of their particulars are simply accidents of evolutionary history. Nevertheless, our work demonstrates that these challenges can be overcome. By curating task design within the limits of deep RL, and designing models with biologically meaningful degrees of freedom, we have demonstrated how to explore the emergence of neural architectures and agent behaviours in accurate and authentic arenas.

## REPRODUCABILITY STATEMENT

All code used to train our agents and run our analyses will be made publicly available at publication time. The state of all trained models used to generate the figures will also be made available. Sufficient details are also provided in the Computational Framework and Results sections to reproduce these models.

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
