# OpenReview forum: "A computational approach to visual ecology with deep reinforcement learning"
_ICLR.cc/2024/Conference — Submitted to ICLR 2024_

### Official Review · Reviewer_bdjp · 2023-10-19

**Soundness:** 3 good
**Presentation:** 3 good
**Contribution:** 3 good
**Rating:** 6
**Confidence:** 3

**Summary:**

The paper trains agents to survive in 3D environments where survival depends on food gathering (and avoidance of harmful items). The appearance of food items can vary in complexity, from two-color items to CIFAR10 classes. Various type of agent architectures are compared, including feedforward vs recurrent, with or without an input for satiety, and linear vs nonlinear activations.

Recurrence and satiety inputs are found to consistently improve performance. However, network size seems to have a very modest effect across conditions, except perhaps for RNNs (judging from Figure 2).

The authors demonstrate that recurrence is used at least in part specifically for object discrimination, and that the agent's learned value function is influenced by both food countdown and satiety.

**Strengths:**

- The experiment is rather interesting in itself.

- The results, such as they are, seem well supported.

- The paper is well written, though some information is missing - see below.

**Weaknesses:**

- The results are not exactly earth-shattering. More difficult tasks seem to benefit from more complex architecture and additional inputs. If anything, the surprising result is the *low* impact of network size on performance (Figure 2).

- Some clarifications are needed, see below.

**Questions:**

- Although the architecture is reasonably well-described, the training itself is not, with little detail except for a mention of PPO. E.g. what is the reward function, exactly?

-  From the Discussion: "In particular, we demonstrated how recurrent brains attended to latent variables beyond the agent’s hunger and the immediate presence of food." - I'm sorry, I missed that part. Can you point out more explicitly where this is shown in the paper and what "latent variables" are "demonstrated" to be attended to, beyond satiety and food items? I agree that this would be quite interesting.

- What's an "ELU" nonlinearity?

---

> ### Author Response · Authors · 2023-11-13
>
> Thank you very much for your positive review.
>
>  We will respond to your criticisms and concern point by point. If you have any further feedback, or believe that implementing certain changes could improve your opinion of our paper, please let us know.
>
> *Weaknesses:*
>
> *The results are not exactly earth-shattering. More difficult tasks seem to benefit from more complex architecture and additional inputs. If anything, the surprising result is the low impact of network size on performance (Figure 2).*
>
> We do think that the devil is in the details here – we find it interesting the way in which architecture and hyperparmeters interact with the task in different ways. For example, the feedforward network is generally insensitive to hyperparameters, whereas the RNN benefits from larger hyperparameters only on the most visually difficult tasks.
>
> *Some clarifications are needed, see below.*
>
> *Questions:*
>
> *Although the architecture is reasonably well-described, the training itself is not, with little detail except for a mention of PPO. E.g. what is the reward function, exactly?*
>
> Although we stated as much in the text, we realize that we explained it poorly: the reward is the current satiety of the agent. We will improve our explanation of the details of the reward and training.
>
> *From the Discussion: "In particular, we demonstrated how recurrent brains attended to latent variables beyond the agent’s hunger and the immediate presence of food." - I'm sorry, I missed that part. Can you point out more explicitly where this is shown in the paper and what "latent variables" are "demonstrated" to be attended to, beyond satiety and food items? I agree that this would be quite interesting.*
>
> This is a fair point. Firstly, the argument we made in the paper was essentially the value function of the recurrent models could not be reduced to satiety and food distance, and therefore it must be sensitive to additional latent variables.
>
> We agree however, that it would be stronger if we could extend our current analysis to actually identify some of these variables. We are currently investigating this and will post any clear results that we find.
>
> *What's an "ELU" nonlinearity?*
>
> An “exponential linear unit”. It is similar to a RELU yet everywhere differentiable.

---

> > ### Comment · Reviewer_bdjp · 2023-11-14
> >
> > Thank you for your reply.
> >
> > > Firstly, the argument we made in the paper was essentially the value function of the recurrent models could not be reduced to satiety and food distance, and therefore it must be sensitive to additional latent variables.
> >
> > Maybe, maybe not. AFAICT, unless such variables are identified and their impact tested, the unexplained variance in V could just as well be due to artifacts of the RNN's dynamics.
> >
> > Currently I am maintaining my evaluation of the paper: mildly interesting study with not-too-surprising results and perhaps some overblown claims (other reviewers have also pointed the tenuousness of the mapping to biological structures). If an updated version is posted I might update my evaluation again.

---

### Official Review · Reviewer_pa3w · 2023-10-27

**Soundness:** 3 good
**Presentation:** 3 good
**Contribution:** 1 poor
**Rating:** 3
**Confidence:** 5

**Summary:**

In this work, the Authors set to study how differences in environments pose different requirements for neural architectures enabling vision and decision-making. To this end, they have implemented a 3D simulation of a foraging task where agents had to collect positively rewarded objects and avoid negatively rewarded objects. The agents comprised of a CNN to process pixel data and an actor-critic RL module for decision-making; they were trained end-to-end using the PPO algorithm. The Authors varied the architectures/inputs of the agents and compared their performance in environments with varied complexity of visual stimuli.

**Strengths:**

The text is well-written and easy to follow.

The description of the experiments is sufficiently detailed.

The question posed by the Authors is interesting and important.

However, the means used to address the question appear insufficient (see below).

**Weaknesses:**

My main concern regarding this work is that the entire study is conducted on a (particular implementation of) an artificial system, only somewhat paralleled to the brain, so all the results may end up being specific to this system and may not generalize beyond it. Should such experiments be conducted with real-life animals representing different “complexities” of neural processing, we would be able to learn something about real-world neurobiology. Should the models explicitly contain and consider particular biologically relevant architectural choices, we could be able to learn about the (focused) impacts of such. In current settings, sadly, the results relevant for a (generic) CNN and PPO may not inform us about such things.

My secondary concern is that I do not agree that end-to-end processing is necessarily beneficial for learning the optimal representations in submodules of the neural network. Specifically, for the longest time, it’s been an ongoing debate with no clear outcome. The examples include audio processing (where SOTA was switching between using mel-spectrograms, then deep-learning representations such as the ones in wav2vec, then again considering mel-spectrograms), vision processing (e.g. using feature-based pre-alignment before training the triplet loss in facial recognition), and control (where control submodules are trained end-to-end, e.g. in quadruped robots with manipulators, but vision models are separately pretrained). Related to this point, I think that the aforementioned cases are relevant and need to be discussed in the paper.

My ternary concern follows that, in this specific work, the end-to-end training may be unnecessary, increasing the training time but, potentially, not providing new insights. In the specific case of foraging, most of the works successfully operate on simple state representations, reducing the computational time from months on a GPU cluster to seconds on a laptop. I am familiar with only one work using 3D simulation for a foraging task but, likewise, I didn’t find their case well-argued. Either way, if there is indeed a benefit of using an end-to-end model in the current framework, it would make sense to highlight this benefit via a baseline analysis where the blocks of the same model are trained separately.

Lastly, if focus on different submodules of the proposed model, the results do not seem surprising. Indeed, more complex architectures are needed to distinguish more complex stimuli (e.g. texture in MNIST digits as opposed to color in apples). Surely, RNNs have the capacity to embody more complete information about the state, compared to feedforward architectures, departing from Markovian task formulation and enabling longer-term planning. These results are known in the respective fields, and the related literature seems relevant enough to be included/discussed in this paper. The other results, such as agents slowing down before stimuli, are interesting but may be specific to the proposed framework – unless proven otherwise.

Overall, while the results are technically correct and well-described, the concerns above sadly preclude me from recommending the paper to be accepted to the ICLR at this point.

**Questions:**

Minor:

-page 2: Merell et al modeled a rat, not a mouse.

-page 3: “the reward at every frame was equal to the current satiety of the agent”. This reward shaping seemingly contradicts an earlier statement that “In our framework we reduced the reward function to the survival of the agent, and avoided further fine tuning of the reward” (page 1).

---

> ### Author Response · Authors · 2023-11-13
> **Addressing Weakness 1**
>
> Thank you for your review. We very much appreciate that despite your fundamental issues with our study, you value the quality of writing and the overall goals of our study. Your critical insights are very helpful.
>
> We will respond to your criticisms and concern point by point. If you have any further feedback, or believe that implementing certain changes could improve your opinion of our paper, please let us know.
>
> *My main concern regarding this work is that the entire study is conducted on a (particular implementation of) an artificial system, only somewhat paralleled to the brain, so all the results may end up being specific to this system and may not generalize beyond it. Should such experiments be conducted with real-life animals representing different “complexities” of neural processing, we would be able to learn something about real-world neurobiology. Should the models explicitly contain and consider particular biologically relevant architectural choices, we could be able to learn about the (focused) impacts of such. In current settings, sadly, the results relevant for a (generic) CNN and PPO may not inform us about such things.*
>
> We believe that there is a fundamental confusion about the purpose of modeling in science. Its purpose is precisely not to make the most realistic model, or always use the real system, but the purpose of modeling is choose the simplest model that will allow you to answer your questions.
>
> Unfortunately, it is clear we did not explain this sufficiently well in the text, but the specification of our vision model is based on established models of early mammalian vision in the computational neuroscience literature (see Lindsey et al (2018) in ICLR, Ocko et al (2018) in NeurIPS, and Maheswaranathan et al (2023) in Neuron). Models with this structure are highly effective at modelling the electrical activity of biological neurons involved in animal vision. In these models the number of channels corresponds to the number distinct biological cell types at each layer of the vision system, which was one of the hyperparameters we explored.
>
> We cited the relevant papers but did not make the connection between our model and existing models of early vision sufficiently explicit. While we did not run neurobiology experiments ourselves, our results provide important insights into existing, data-driven models of sensory neural circuits, by generalizing their application from simple classification tasks to model ecologies. In fact, much of the field of representation learning has its foundations in this kind of work, going back to Barlow and colleagues in the 1950s and 1960s. In future work we would of course like to gather data and make the model more complex, but we believe the scope of our paper is sufficient for publication.
>
> Finally, even if sample inefficient, PPO appears sufficient for our aims.

---

> ### Author Response · Authors · 2023-11-13
> **Adressing Weaknesses 2 and 3**
>
> *My secondary concern is that I do not agree that end-to-end processing is necessarily beneficial for learning the optimal representations in submodules of the neural network. Specifically, for the longest time, it’s been an ongoing debate with no clear outcome. The examples include audio processing (where SOTA was switching between using mel-spectrograms, then deep-learning representations such as the ones in wav2vec, then again considering mel-spectrograms), vision processing (e.g. using feature-based pre-alignment before training the triplet loss in facial recognition), and control (where control submodules are trained end-to-end, e.g. in quadruped robots with manipulators, but vision models are separately pretrained). Related to this point, I think that the aforementioned cases are relevant and need to be discussed in the paper.*
>
> *My ternary concern follows that, in this specific work, the end-to-end training may be unnecessary, increasing the training time but, potentially, not providing new insights. In the specific case of foraging, most of the works successfully operate on simple state representations, reducing the computational time from months on a GPU cluster to seconds on a laptop. I am familiar with only one work using 3D simulation for a foraging task but, likewise, I didn’t find their case well-argued. Either way, if there is indeed a benefit of using an end-to-end model in the current framework, it would make sense to highlight this benefit via a baseline analysis where the blocks of the same model are trained separately.*
>
> This is a very important point, and one we should have better addressed in the paper. Firstly, we must reiterate that the goal of our simulations was not simply to find the agent that achieves the highest performance, but rather to understand the conditions under which various behaviours and representations emerge in an agent.
>
> The actual RL problem is easy by design, and we do not doubt that it could be solved by a clever combination of pretrained vision model and simpler RL training. In fact, our existing results show this: in the apples task, where there are only two visually distinct objects that are entirely colour separated, agents can more or less maximize their performance after a couple hours of training. Using an appropriately pre-trained vision model should afford similarly fast learning on the other tasks, because it is only the visual statistics that are changing from task to task.
>
> Nevertheless, we believe end-to-end training is critical to maintaining the ecological validity of our approach for at least two reasons: (i) We don’t want to train the agent on anything it might not experience in its ecological niche, and (ii) we want to ensure that the agent only learns what is relevant for the task, and ignores anything else. Pretraining the vision model to e.g. reconstruct a large image dataset would violate these principles.
>
> We have a large set of experiments that we cut from the paper due to space constraints: we generated a set of “distractors” tasks, where the environment was filled with e.g. apples and cifar images, where the apples images impacted satiety and cifar did not, and vise versa. What we found was that in terms of how the agent learned and its ability to identify food, the distractors had no effect on the agent. In other words, an agent that only gets nourished by apples does not learn to differentiate cifar images from each other because they are irrelevant. This notion of survival relevance is one of our key interests in our framework, and would not be captured by modular training where the objects of the vision model and policy were separated.
>
> If you would feel that these results might significantly improve the paper, we could rework the paper to include them.

---

> ### Author Response · Authors · 2023-11-13
> **Adressing Weaknes 4 and Questions**
>
> *Lastly, if focus on different submodules of the proposed model, the results do not seem surprising. Indeed, more complex architectures are needed to distinguish more complex stimuli (e.g. texture in MNIST digits as opposed to color in apples). Surely, RNNs have the capacity to embody more complete information about the state, compared to feedforward architectures, departing from Markovian task formulation and enabling longer-term planning. These results are known in the respective fields, and the related literature seems relevant enough to be included/discussed in this paper. The other results, such as agents slowing down before stimuli, are interesting but may be specific to the proposed framework – unless proven otherwise.*
>
> We will endeavour to better cite relevant literature when the results of our simulations align with known results.
>
> One of our results that you did not mention is that an RNN also seems essential for solving the recognition problem on the CIFAR-10 task. We believe that we showed this and other results with a high degree of rigour – we would very much appreciate if you could explain how we might prove our results outside of the framework in which we test them or point us to relevant references.
>
> *Overall, while the results are technically correct and well-described, the concerns above sadly preclude me from recommending the paper to be accepted to the ICLR at this point.*
>
> Although we would certainly prefer for you to recommend our paper for acceptance, we nevertheless very much appreciate your thorough and thoughtful criticisms.
>
> *Questions:*
>
> *-page 2: Merell et al modeled a rat, not a mouse.*
>
> Thank you for pointing this out.
>
> *-page 3: “the reward at every frame was equal to the current satiety of the agent”. This reward shaping seemingly contradicts an earlier statement that “In our framework we reduced the reward function to the survival of the agent, and avoided further fine tuning of the reward” (page 1).*
>
> Yes this was unclear in our text. We will fix this.

---

> > ### Comment · Reviewer_pa3w · 2023-11-20
> >
> > My apologies for the delayed responses. I wanted to take as much time as possible to read through the discussion here and through the paper again in an attempt to embrace the Authors' viewpoint.
> >
> > Throughout the paper and the discussion here, the Authors clarify that the goal of the paper was to understand how visual ecology shapes animal vision. That's surely an important goal. Further, the Authors elaborate that they've used the end-to-end architecture because of their goal, i.e. to analyze visual processing and visual representations that emerge to solve the task at hand. The visual representations, it follows, emerge at the interface of the vision and control modules, so both have to be explicitly trained within the model. Sounds reasonable. For the vision processing, the Authors use the model from several prior papers building the links between the primate visual system and the CNNs. For the model, they use as simple of a foraging paradigm as possible to observe differences in visual processing while not adding unnecessary additional complexity, aiming at an appropriate level of abstraction, as any model should.
> >
> > While it's hard to disagree with the high-level logic and the general arguments presented in the paper and clarified here in the responses, the devil -- as acknowledged by the Authors -- is in details.
> >
> > The model complexity surely should reflect the question at hand, but is the proposed model adequate to answer this question? In this work, the question of "how visual ecology shapes animal vision" was transformed to "how a randomly initialized visual system would learn tasks of varied complexity". While the learning from the "randomly initialized visual system" arguably happens over evolutionary timescales, such learning most likely includes a multitude of different tasks, aiming at generalist representations that can be reused in different tasks. At the same time, the optimal connectivity has to be passed to the next generations through the "genomic bottleneck" thus imposing additional constraints on the low-dimensional structure of connectivity. During the life of a single animal, these constraints result in the visual system being pre-wired in a certain way, so a randomly initialized network may not reflect the initial conditions for learning in individual animals. For these reasons, the way in which the (valid) research question here has been reformulated into the model does not appear adequate to enable answering that original question.
> >
> > Second, the visual system model was borrowed from prior literature. As the Authors have correctly pointed out, different models may be adequate for answering different questions by faithfully capturing relevant features of the phenomenon at hand. As a corollary, while representing the visual system as a CNN may have faithfully served the purposes of the aforementioned papers, there is no a priori reason to expect that such a relevance would transfer to this paper pursuing its own scientific question. Moreover, chances are that the models are inadequate even for the aims pursued in some prior research. The most proper way to address this issue would be to convincingly explain why a certain CNN architecture is good for modeling the specific task at hand and then point the reader to the prior literature that may have already used similar logic in related tasks.
> >
> > Third, the findings here describe a single model architecture trained under a certain choice of parameters. How are we supposed to know that the results obtained here are not specific to these choices rather than to the system under study? Different weight initializations, hyperparameter choices, optimizers, learning rates, and training protocols are known for their ability to produce drastically different results. To address that issue, one needs to either go through different parameter choices to show that the results hold or justify a specific choice of parameters by additional arguments relating these parameters to the modeled system. An even better way to get more insights would be to have contrasting models, all biologically plausible, and see the contrasting predictions that then could be compared to the data such that one could be ruled out and we could learn something about the system at hand.
> >
> > Finally, as noted by most of the Reviewers, the results overall are not surprising to the machine learning community, as in a more complex task necessitates a more complex model. Therefore, as it currently stands, the biological relevance of the work needs to be further proven, and machine learning results of the work are expected. This is due to the implementation choices, even though I wholeheartedly agree with the Authors' overall logic. As a result, at this point, my valuation of the manuscript stands the same.

---

### Official Review · Reviewer_PwAE · 2023-10-31

**Soundness:** 3 good
**Presentation:** 3 good
**Contribution:** 2 fair
**Rating:** 5
**Confidence:** 3

**Summary:**

This paper develops a simple environment off of ViZDoom that studies the problem of how neural net architectural choices affect performance of varying-complexity visual tasks. Authors develop 4 visual tasks that each require a different level of discriminative image processing capacity to perform well. Each task has objects, which are all placed as vertical images atop a 3D-pixelated plane, and images are classified into 10 different satiety scores, {-25, -20, ..., -5, 10, 20, ..., 50}. All tasks involve choosing an action at each timestep (forward/stationary/backward) in a specific direction (left, right, center) to avoid negative-satiety-score objects and make contact with positive-satiety-score objects to prevent satiety from falling to 0, which kills the agent. The paper calls this “visual ecology.”

Authors analyze the effect of architecture and hyperparameters on the lifespan of the agent in each of the 4 tasks, finding a loose correlation between larger kernel count and improved performance. RNNs performed better than fully connected networks on image embeddings, and inputting the satiety score allows the learned value function to more stably track the actual satiety value.

**Strengths:**

- An interesting problem to study: how environments affect visual processing.
- Did a thorough set of experiments given the environment’s capabilities.
- Writing mostly made sense and was clear throughout.

**Weaknesses:**

### Overall
(A1) Paper overall seems geared to a slightly less technical audience than ICLR. For instance, describing each conv layer as a different region of the brain (Section 2.1) seems like a weird thing to do, at least in the AI community. Why is the 4th conv layer analogized to the lateral geniculate nucleus, for example? What is the basis for creating these mappings between conv layers and mammalian brain regions?

(A2) Additionally, some results are presented as interesting by the authors, but to me seem relatively straightforward and expected. For instance, including the input satiety into the architecture should obviously improve performance.

(A3) Related work section should be much more substantial. Have prior papers studied “visual ecology” or done analysis on neural vision architectures when given an environment the agent must survive/do tasks in? It is hard to evaluate the contributions of this paper without a solid summary of previous work in this area.

### This paper is mainly limited in its analysis by an overly simple environment design.
(B1) Based on Figure 4C, it seems like satiety decreases at a constant rate, no matter what action the agent is taking. This seems undesirable, as animals expend different amounts of energy depending on different actions. For instance, being stationary should be much less draining on satiety than moving.

(B2) Environment lacks interesting actions beyond moving in different directions or staying put. Additional simple actions may make the environment much more interesting, such as rotating field of view without changing (x,y) position, and needing to perform some manipulative action (such as triggering a set of discrete actions, such as “pulling vegetables out of the ground,” before being able to consume them). These additional actions can also have different satiety costs to them.

(B3) Environment does not accurately model effects of satiety on actions. Decreased satiety should have a harmful effect on the agent being able to take actions. A satiety=1 agent should not be as effective at moving around as a satiety=100 agent. Under this situation, it would be cool to analyze the satiety value at which the agent is most effective at finding food, since it would tend to be more stationary at higher satieties and tend to be less effective at moving around at lower satieties.

### The main claims in the paper could be better analyzed and argued.
(C1) First abstract claim: “The complexity of the vision model required for survival on this task scaled with the variety and visual complexity of the food in the environment.” This may sound trivial, but authors should define model complexity. Is model complexity only dependent on parameter count? That seems inadequate, since a huge, deep feed-forward network with the same parameter count as a CNN or an RNN should still not be as “complex” due to the inductive biases encoded in the latter two networks. Some of the author’s experimental choices for looking at complexity, such as focusing a lot on number of channels, is probably misguided, since increasing number of channels after a point is known to saturate network performance, as Figures 2D-F show. Things like residual connections, kernel dimension/size, and number of conv layers will probably make for more interesting graphs.

(C2) Second abstract claim: “recurrent network architecture was necessary...for visually demanding tasks.” This claim makes sense from Figure 2C, but I do not feel like it is well-substantiated. For instance, one could feed the current image as well as the last $k-1$ images into the CNN, either stacked channel-wise or arranged as an $(m, n)$ array of images, such that $k = m\times n$. This would not involve the network being recurrent, but still captures information in the previous $k-1$ observations, and it is possible that this does comparably to RNNs. This claim also might not hold if transformers were used as the architecture.

(C3) Third abstract claim: “Different network architectures learn distinct representations of environment and task, leading to different behavioral strategies.” Agent behavior was better investigated in the results, but not necessary the “distinct representations” part of this claim, though there was Figure 4C which showed the different value functions. One suggestion here is that it would be better to revise Figure 4B and 4C, for instance, to show the sensitivity of all 4 methods of $\hat{V}$ on the same image observation at the same location in the environment, so that readers can see the “distinct representations” in image space as well as in value space.

**Questions:**

1. How would behavior have changed if there were no drive for survival, but just a drive for collecting high reward (without a limit of 100)?
2. Suggestion: Authors should compare their designed environment with Fruitbot in ProcGen (https://github.com/openai/procgen#environments), where an agent also tries to get good objects and avoid bad objects.
3. What was the motivation behind choosing Gabors as one of the tasks? Visually, it seems to be the most contrived of the 4 tasks.
4. Are object positions in the environment randomly initialized?
5. What was the mean distance (in terms of optimal number of actions to reach) between each positive-satiety object and its closest positive-satiety neighbor? What was the actual distance traveled by the agent? May be good to measure this. This would be similar to Figure 6’s “Wasted Nourishment” but would instead be measuring “Wasted actions.”
6. Was each environment initialized to have an equal frequency of positive and negative satiety objects?
7. How was the satiety inputted into the network? Was it normalized to be a value between (0, 1)? One could try inputting the satiety into the CNN directly with FiLM layers (https://arxiv.org/pdf/1709.07871.pdf) and see if this increases the effect of input satiety over non-input satiety architectures. Also, why was the input satiety concatenated for the second FC layer instead of the first?
8. Precisely define the reward function. Section 1 says it is “the survival of the agent,” but does that mean it is a sparse 0/1 reward, or is it a constant +1 for all timesteps the agent is alive, and 0 else?

---

> ### Author Response · Authors · 2023-11-13
> **Addressing Overall Weaknesses**
>
> Firstly, I would like to thank you for your extremely thorough review. This insights will help us improve our work.
>
> We will respond to your criticisms and concern point by point. If you have any further feedback, or believe that implementing certain changes could improve your opinion of our paper, please let us know.
>
> *(A1) Paper overall seems geared to a slightly less technical audience than ICLR. For instance, describing each conv layer as a different region of the brain (Section 2.1) seems like a weird thing to do, at least in the AI community. Why is the 4th conv layer analogized to the lateral geniculate nucleus, for example? What is the basis for creating these mappings between conv layers and mammalian brain regions?*
>
> Thank you for pointing this out, as we did not make this sufficiently clear in our writing. The goal of this paper is not to provide technical innovations of deep RL, but rather to understand how visual ecology shapes animal vision. The specification of our vision model is based on established models of early mammalian vision in the computational neuroscience literature (see Lindsey et al (2018) in ICLR, Ocko et al (2018) in NeurIPS, and Maheswaranathan et al (2023) in Neuron) - we cited these papers in our text but did not make the connection between our model and existing models explicit.
>
> Models with the structure of our vision model are highly effective at modelling the early visual system. In a simplified ANN model of the early visual system, the LGN can be understood as a 4th layer of neurons after the three layers of the retina (see Lindsey et al (2018)).
>
> *(A2) Additionally, some results are presented as interesting by the authors, but to me seem relatively straightforward and expected. For instance, including the input satiety into the architecture should obviously improve performance.*
>
> It is of course not surprising that providing additional information to the agent could lead it to better performance, yet we believe interesting findings emerged from studying this architecture. Firstly, adding this signal improved agent performance by unlocking qualitatively distinct behaviours, for example waiting to be sufficiently hungry before eating food. Secondly, even though we showed that RNN-based architectures could estimate their own satiety, they did not exhibit this behavioural strategy. In this context, it seems worth reporting the degree to which this strategy improved agent performance.
>
> *(A3) Related work section should be much more substantial. Have prior papers studied “visual ecology” or done analysis on neural vision architectures when given an environment the agent must survive/do tasks in? It is hard to evaluate the contributions of this paper without a solid summary of previous work in this area.*
>
> This is a fair point, and we apologize for insufficiently summarizing the existing literature. To the best of our knowledge this study is the first of its kind that studies neural network models of visual circuits in simulated ecologies where the agent must survive. We will improve the related work section to better situate our paper.

---

> ### Author Response · Authors · 2023-11-13
> **Addressing Criticisms of Environmental Simplicity**
>
> *This paper is mainly limited in its analysis by an overly simple environment design.*
>
> We agree that the environment is simple. However, we are convinced that this type of environment is the simplest possible that teaches us something interesting on how the statistics of the environment and behavioural goals interact to drive the evolution of neural representations, and choosing the simplest possible model to understand a phenomenon is at the heart of science. We are excited to explore more complex environments in future work, however, adding additional complexities to the environment can also easily obscure any potential findings by introducing confounding variables.
>
> *(B1) Based on Figure 4C, it seems like satiety decreases at a constant rate, no matter what action the agent is taking. This seems undesirable, as animals expend different amounts of energy depending on different actions. For instance, being stationary should be much less draining on satiety than moving.*
>
> We agree that this would be an interesting manipulation. Of course, there are a large variety of innovations one could build into the model, and the one you mention is only one of them. One that we explored was whether an animal has access to its satiety state. Others include multiple agents, competing behavioral goals and so on. Any of these would be interesting to study in the future.
>
> *(B2) Environment lacks interesting actions beyond moving in different directions or staying put. Additional simple actions may make the environment much more interesting, such as rotating field of view without changing (x,y) position, and needing to perform some manipulative action (such as triggering a set of discrete actions, such as “pulling vegetables out of the ground,” before being able to consume them). These additional actions can also have different satiety costs to them.*
>
> In fact, it is possible to the agent to stay put and rotate its visual field, and trained agents do indeed exhibit this behaviour. We will explain more clearly in a revised text. Regarding your second point, the goal of this study is not to identify the most difficult task we can get an agent to perform, but to understand how visual ecology shapes the architecture of animal vision. If we simply make the task as difficult as possible, then we won’t be able to distinguish between poor performance due to the vision model/brain of the agent, and poor performance due to the difficulty of the task. Adding these task innovations would only be reasonable if we have a clear idea of the conditions under which an agent can solve the simplest task, or how “pulling vegetables out of the ground” creates a new constraint on the visual system that may influence the learned representation.
>
> *(B3) Environment does not accurately model effects of satiety on actions. Decreased satiety should have a harmful effect on the agent being able to take actions. A satiety=1 agent should not be as effective at moving around as a satiety=100 agent. Under this situation, it would be cool to analyze the satiety value at which the agent is most effective at finding food, since it would tend to be more stationary at higher satieties and tend to be less effective at moving around at lower satieties.*
>
> Again, we very much appreciate the intuition behind this suggestion, and adding innovations like this would be appropriate for future work. However, we see no obvious reason why adding this additional complexity is necessary for the quality of our results, and otherwise keeping the environment as simple as possible is an advantage of our current study.

---

> ### Author Response · Authors · 2023-11-13
> **Addressing Insufficient Analyses**
>
> *The main claims in the paper could be better analyzed and argued.*
>
> *(C1) First abstract claim: “The complexity of the vision model required for survival on this task scaled with the variety and visual complexity of the food in the environment.” This may sound trivial, but authors should define model complexity. Is model complexity only dependent on parameter count? That seems inadequate, since a huge, deep feed-forward network with the same parameter count as a CNN or an RNN should still not be as “complex” due to the inductive biases encoded in the latter two networks. Some of the author’s experimental choices for looking at complexity, such as focusing a lot on number of channels, is probably misguided, since increasing number of channels after a point is known to saturate network performance, as Figures 2D-F show. Things like residual connections, kernel dimension/size, and number of conv layers will probably make for more interesting graphs.*
>
> This is all quite fair. Firstly, we do mostly use “complexity” as a shorthand for parameter count. We will clarify this in the text.
>
> Secondly, there are biologically motivated arguments for varying some of the architectural features as you suggest, and we would like to explore them. Nevertheless, we focused on channel count for the reasons mentioned above (which again we did not explain sufficiently well in the text). We would also point out that performance does not saturate based on the number of channels on the CIFAR-10 task for the RNN network, which was one of the effects we were happy to isolate.
>
> *(C2) Second abstract claim: “recurrent network architecture was necessary...for visually demanding tasks.” This claim makes sense from Figure 2C, but I do not feel like it is well-substantiated. For instance, one could feed the current image as well as the last k-1 images into the CNN, either stacked channel-wise or arranged as an array of images, such that k = m x n. This would not involve the network being recurrent, but still captures information in the previous k-1 observations, and it is possible that this does comparably to RNNs. This claim also might not hold if transformers were used as the architecture.*
>
> This is again a fair point and could well be true. However, it’s hard to imagine that this architecture would be biologically plausible – this would correspond to something like maintaining a buffer of activity in the retina of the agent. We appreciate that the term “biologically plausible” can be vague, but it’s also not without meaning, and there are a lot of architectures we can rule out based on existing understanding and models of animal vision.
>
> *(C3) Third abstract claim: “Different network architectures learn distinct representations of environment and task, leading to different behavioral strategies.” Agent behavior was better investigated in the results, but not necessary the “distinct representations” part of this claim, though there was Figure 4C which showed the different value functions. One suggestion here is that it would be better to revise Figure 4B and 4C, for instance, to show the sensitivity of all 4 methods of \hat{V} on the same image observation at the same location in the environment, so that readers can see the “distinct representations” in image space as well as in value space.*
>
> Again, this is a fair point, and it is something we considered, but there are challenges to doing this rigorously. The main issue is that recurrent networks would require the recent history of images to be loaded as well, but then the history would be based on the policy of one agent architecture, i.e. the network with or without satiety. We know that networks with/without satiety behave quite differently, and so we can’t rule out that e.g. feeding an RNN that includes input satiety (IS) with an image history generated by an RNN without input satiety would lead the IS-RNN to generate uncharacteristic values of \hat{V}. Nevertheless, if this seems like a secondary concern to the reviewer, we could easily add this to a revised version of the manuscript – perhaps to the supplement at least.

---

> ### Author Response · Authors · 2023-11-13
> **Addressing Questions**
>
> *How would behavior have changed if there were no drive for survival, but just a drive for collecting high reward (without a limit of 100)?*
>
> This would be worth exploring further, though arguably it’s beyond the scope of our manuscript. Nevertheless, we did design our task to exert a high degree of time pressure on the agent. Removing this would likely make the CIFAR-10 agents perform much better, as they could spend more time ensuring they recognize the images correctly.
>
> *Suggestion: Authors should compare their designed environment with Fruitbot in ProcGen (https://github.com/openai/procgen#environments), where an agent also tries to get good objects and avoid bad objects.*
>
> Thank you, this is a very helpful reference, and we will cite it in our text.
>
> *What was the motivation behind choosing Gabors as one of the tasks? Visually, it seems to be the most contrived of the 4 tasks.*
>
> Our goal was to define a simple dataset with a small number of images, that would still apparently require non-linear computation to identify – in comparison with the red/blue apples task, the model can’t simply take advantage of the colour channels to separate the black and white Gabors. Based on the results we present in Figure 2, the Gabors datasets achieves its purpose. Gabors also have a long history in visual neuroscience, and so serves as a point of reference for some of our readership.
>
> *Are object positions in the environment randomly initialized?*
>
> Yes. The task would be much easier otherwise. We will clarify this in the text.
>
> *What was the mean distance (in terms of optimal number of actions to reach) between each positive-satiety object and its closest positive-satiety neighbor? What was the actual distance traveled by the agent? May be good to measure this. This would be similar to Figure 6’s “Wasted Nourishment” but would instead be measuring “Wasted actions.*
>
> We will measure this number and include it.
>
> *Was each environment initialized to have an equal frequency of positive and negative satiety objects?*
>
> The environment was initialized with more positive than negative satiety objects. This helped agents avoid a local minimum where they refused to engage with the task because it is initially too painful.
>
> *How was the satiety inputted into the network? Was it normalized to be a value between (0, 1)? One could try inputting the satiety into the CNN directly with FiLM layers (https://arxiv.org/pdf/1709.07871.pdf) and see if this increases the effect of input satiety over non-input satiety architectures. Also, why was the input satiety concatenated for the second FC layer instead of the first?*
>
> Satiety was indeed normalized between 0 and 1. Thank you for the reference, we will look into it.
>
> We input the satiety at the second layer, because we reason it would be helpful to have an FC layer which processes the images without input satiety. We did not thoroughly investigate variations of this.
>
> *Precisely define the reward function. Section 1 says it is “the survival of the agent,” but does that mean it is a sparse 0/1 reward, or is it a constant +1 for all timesteps the agent is alive, and 0 else?*
>
> As stated in the text the reward is the current satiety of the agent, however we see now how this might be confusing considering our statements about reward being “only the survival” of the agent. We will clarify this in the text.
>
> ---
>
> I would like to thank you once again for your thorough and comprehensive review.

---

### Official Review · Reviewer_zf5P · 2023-11-03

**Soundness:** 2 fair
**Presentation:** 2 fair
**Contribution:** 2 fair
**Rating:** 3
**Confidence:** 4

**Summary:**

Motivated by the evolution of animal vision, this paper trains agents in a foraging task with deep RL, in which the complexity of the tasks is varied based on the visual complexity of the food images. The most complex food representations were based on CIFAR-10 images. In contrast to many other deep RL domains, agents are only rewarded for surviving. The results show that more complex visual complexity requires more complex vision models. An interesting observation is that for most complex tasks, recurrent network architectures were necessary. Additionally, the authors show that different network architectures learn different representations of the environment.

**Strengths:**

- Interesting idea to use recent advances in DNN to study visual ecology
- Since most neural networks for image recognition are purely feedforward it is an interesting result that recurrence facilitates object discrimination on visually complex tasks How is it using the recurrent dynamics?
- Section 3.3. in particular presents an interesting investigation into how the neural network architecture shapes the reward system of the agent

**Weaknesses:**

- My main issue is the comparison of a simple CNN to an animal vision system, i.e. "We modeled the CNN after the early mammalian visual system: the base layers were grouped sequentially into the photoreceptor (PR), bipolar (BP), retinal ganglion cell (RGC), lateral geniculate nucleus (LGN), and primary visual cortex (V1) layers.” As far as I understand, it’s just a different number of channels and kernel sizes? Naming the different layers in a network after biological brain regions does not directly make them more biologically realistic.
- Environments in nature are much more complex than the ones proposed in this paper. To study visual ecology, it seems our agent environments need to be more complex as well.
- There is a lot of related work in the evolutionary community, which isn’t mentioned, where survival is the only reward mechanism.
- In conclusion, I would argue that a computational framework on visual ecology has to go beyond one experiment/domain with a slight variation on a deep RL setup

**Questions:**

- How would a model perform that is not "based on" the mammalian visual system?
- How realistic is the vision model when compared to animal vision?
- In nature, adaptation is a result of evolution and lifetime learning. Wouldn’t it be important for this research to combine both adaptation mechanisms in some way? In particular, the evolution of plastic neural networks with Hebbian learning rules could be relevant here.

---

> ### Author Response · Authors · 2023-11-13
> **Addressing Weaknesses**
>
> Firstly, I would like to thank you for your review, and identifying some of the strengths of our paper. We will respond to your criticisms and concern point by point. If you have any further feedback, or believe that implementing certain changes could improve your opinion of our paper, please let us know.
>
> *My main issue is the comparison of a simple CNN to an animal vision system, i.e. "We modeled the CNN after the early mammalian visual system: the base layers were grouped sequentially into the photoreceptor (PR), bipolar (BP), retinal ganglion cell (RGC), lateral geniculate nucleus (LGN), and primary visual cortex (V1) layers.” As far as I understand, it’s just a different number of channels and kernel sizes? Naming the different layers in a network after biological brain regions does not directly make them more biologically realistic.*
>
> This is a fair point, and we could have explained better. The specification of the vision model of our RL agent is based on established models of early mammalian vision in the computational neuroscience literature (see Lindsey et al (2018) in ICLR, Ocko et al (2018) in NeurIPS, and Maheswaranathan et al (2023) in Neuron) - we cited these papers in our text but did not make the connection between our model and existing models explicit. Of course, such models are highly simplified, but the patterns of convergence and divergence of channels are modelled after what is happening in the visual system. In fact, such models are highly effective at modelling the activity of biological neurons in the early visual system (Maheswaranathan et al (2023)). So, while simple, the model is useful to our goal of uncovering the principles by which the environment and behavioral goals shape neural representations.
>
> *Environments in nature are much more complex than the ones proposed in this paper. To study visual ecology, it seems our agent environments need to be more complex as well.*
>
> We agree that the world is indeed much more complex than the simple environment we use. However, we are convinced that this type of environment is the simplest possible that teaches us something interesting on how the statistics of the environment and behavioral goals interact to drive the evolution of neural representations, and choosing the simplest possible model to understand a phenomenon is at the heart of science. For example, if we make the environment or the task more complex, it becomes much more difficult to distinguish performance differences that were due to the difficulty of learning the task, versus differences that were due to the limitations of the vision system. For the purposes of this first study, we therefore found it important to keep the former as simple as possible, towards the goal of understanding the latter. In future work we indeed hope to study how agents learn and behave in more complex environments.
>
> *There is a lot of related work in the evolutionary community, which isn’t mentioned, where survival is the only reward mechanism.*
>
> This is a fair point. We will  include citations from the evolutionary community to better motivate this part of our framework. We would certainly appreciate any recommendations you might have.
>
> *In conclusion, I would argue that a computational framework on visual ecology has to go beyond one experiment/domain with a slight variation on a deep RL setup*
>
> As explained above, the goal of our work is to extract the principles by which an agent’s visual system is shaped by the environment and its behavioral affordances. To the best of our knowledge, this is not something that deep RL has been used for. Furthermore, our computational framework is capable of supporting a wide variety of experiments and domains, and a wide variety of RL setups. Also, given (i) the amount of compute it takes to run deep RL simulations, and (ii) the unbounded number of experiments that might constitute all possible visual ecologies, we would argue that it is important to rigourously understand a base case as well as possible, against which more complicated cases may then be compared.

---

> > ### Comment · Reviewer_zf5P · 2023-11-20
> >
> > Thank you for the clarifications. However, I still very much agree with the response of reviewer pa3w who raises some important issues about the details of the comparisons in the paper. I believe this is a very interesting direction for future work (and I'm looking forward to seeing what the authors will do next with this framework) but in its current state, I do not think it's quite ready yet for the main conference.

---

> ### Author Response · Authors · 2023-11-13
> **Addressing Questions**
>
> *Questions:*
>
> *How would a model perform that is not "based on" the mammalian visual system?*
>
> This is an interesting question, although we wouldn’t quite know how to start answering it, because CNNs are based on the mammalian visual system. Perhaps something based on transformers, although we suspect a thorough characterization would be beyond the scope of one paper.
>
> *How realistic is the vision model when compared to animal vision?*
>
> There are two answers to your question: One the one hand, the model is more realistic than it might appear - please see our response above. On the other hand, it is quite simplistic, but for good reasons, as also explained above. The question should be: is our model able to answer interesting questions: And we think it does.
>
> *In nature, adaptation is a result of evolution and lifetime learning. Wouldn’t it be important for this research to combine both adaptation mechanisms in some way? In particular, the evolution of plastic neural networks with Hebbian learning rules could be relevant here.*
>
> Thank you for bringing this up. Currently, the two types of learning are somewhat conflated in our setup and we will make this more clear in the Discussion. In the future, we might explore an evolutionary approach to modifying the architecture of the network. With regards to Hebbian Learning, we are constrained by what can be efficiently trained in the context of a Deep RL system, and so for the purposes of this study we relied on conventional architectures that should in principle perform well.

---

### Meta-Review · Area_Chair_8Y1X · 2023-12-13

**Metareview:**

The paper describes a test environment based on VizDoom and CIFAR-10 where agents are rewarded simply for surviving.

The main strength is the setup of the experiment itself, which is clever and novel.

The main weaknesses are that (1) we don't actually learn that much, in particular nothing we could not have guessed beforehand, and (2) it is not clear that what we learn would generalize to other settings and problems.

**Justification For Why Not Higher Score:**

The results are neither strong nor surprising, and the generalizability is unclear.

**Justification For Why Not Lower Score:**

N/A

---

### Decision · Program_Chairs · 2024-01-16

Reject